# Effects of tai chi on postural balance and quality of life among the elderly with gait disorders: A systematic review

Fan Xu[1]☯*, Kim Geok Soh[1]☯, Yoke Mun Chan[2]‡, Xiao Rong Bai[3]‡, Fengmeng Qi[1]‡, Nuannuan Deng[1]‡

1 Faculty of Education Studies, Department of Sport Studies, Universiti Putra Malaysia, Selangor, Malaysia, 2 Faculty of Medicine and Health Sciences, Department of Dietetics, Universiti Putra Malaysia, Selangor, Malaysia, 3 Faculty of Sports Studies, Huzhou University, Huzhou, China

☯ These authors contributed equally to this work.
‡ These authors also contributed equally to this work.
* gs62396@student.upm.edu.my

**Data Availability Statement:** All relevant data are within the manuscript and its Supporting Information files.

## Abstract

### Background

Tai Chi is good for improving the physical fitness of older adults. But few studies have reported the effects of Tai Chi on the postural balance and quality of life of older adults with gait disorders.

### Objective

This review aimed to assess the influence of tai chi on postural stability and quality of life in older adults with abnormal gait.

### Method

According to the literature retrieval principles, the works published from the inception date to May 2023 were retrieved, including the following databases: PubMed, Scopus, Web of Science, China National Knowledge Infrastructure, EBSCOhost, and Google Scholar. Subsequently, literature screening and quality assessment were performed.

### Results

A total of 16 randomized controlled trials were included in this study, Tai Chi intervention can affect populations with Parkinson's disease (PD), no exercise, mild cognitive impairment (MCI), chronic stroke, sedentary, fear of falling, or history of falling. Postural instability is associated with balance, gait, the Unified Parkinson's Disease Rating Scale Motor Subscale 3 (UPDRS III), mobility, lower body strength, and falls. Only two articles looked at quality of life. The Yang style is the most commonly used in the intervention. Nonetheless, most studies were performed on female participants, hence, more research on older male populations is needed.

**Funding:** The author(s) received no specific funding for this work.

**Competing interests:** The authors have declared that no competing interests exist.

## Conclusion

Tai Chi intervention benefits postural balance in patients with gait disorders. 12 weeks is the most common intervention period for patients with gait disorders. The frequency of intervention is seven articles twice a week, and the intervention time is about 60 minutes. The Tai Chi intervention methods in this study involve Yang Style, Sun Style, Taoist Tai Chi, and Health Qigong Tai Chi, but the Yang Style Tai Chi intervention is the most widely used.

## 1 Introduction

In recent years, the aging population has risen tremendously in many countries than in the past. According to the United Nations (2019), one-sixth of the world's population will be over 60 years old by 2030 [1]. Consequently, health issues among the elderly have become a global concern, particularly their general well-being and increased vulnerability to diseases. Therefore, there is a massive demand for medical and social services, such as preventive rehabilitation [2]. For example, up to 75% of residents aged $\geq$ 60 in China suffer from chronic diseases [3]. Furthermore, the physiological and age factors delay functional recovery among the elderly, hence, prone to injury [4]. Currently, the leading cause of injury or death among elderlies > 65 is falling, thus, a major concern for those in this age group [5]. Furthermore, social isolation and depression among older adults also increased their risk of falling [6]. Apart from that, biological causes such as poor balance and flexibility, sensory impairment, and low muscle strength can contribute to falling in this age group [6]. Thus, balance improvement is essential to maintain their walking ability and reduce fall rates [7,8].

Exercise as a complementary therapy could improve neuromuscular function, but monotonous technique or strength training is not suitable for all patients. For example, for some older adults, many exercise activities are either too intense or too monotonous, which makes them unable to persist [9]. Therefore, it is crucial to determine the best fitness routine for different age groups, particularly the elderly. As an aerobic exercise with a long history in China, tai chi is easy to learn, effective, and safe due to the medium-to-low intensity; hence, suitable for the elderly [10]. Furthermore, Tai Chi, the most popular Chinese martial art originated in ancient China, involves physical and mental exercises [11]. Contrary to the general Chinese martial arts, Tai Chi is an art that undergoes continuous development based on ancient Chinese philosophy. Thus, new styles have been created through the unique insights of Tai Chi practitioners for generations [12] while maintaining softness, slowness, and rhythmic features [13]. A growing body of research shows that tai chi is beneficial for older generations. For instance, a study found that practicing traditional Chinese sports can improve gait balance and increase body strength [14,15]. In addition, this intervention is an effective therapy for improving postural balance in older adults.

Postural disturbances often lead to imbalance and fall, especially among older adults [16]. Furthermore, the incidence of gait disorders increases with age; approximately 10% occur in people aged 60 to 69 and about 60% for those > 80 years old [17]. Gait disorders among older adults are generally difficult to detect because the older adults symptoms are subtle. Manifestations of gait disorders in older adults patients are likely to be masked by regular aging-related physiological changes, leading to the complexity of the condition [18]. Notably, individuals with neurological disorders often experience gait and balance disturbances that affect their quality of life (QoL) [19]. As the second most common neurodegenerative disease, Parkinson's disease is a threat to the elderly population worldwide [20,21], which mainly manifests as a

dysfunction of the body's motor system. For example, patients often face problems such as slow movement, unstable posture, and abnormal gait [22,23]. Additionally, stroke impacts gait flexibility and mobility; thus, most survivors lose their ability to walk independently.

Long-term lack of exercise among older adults accelerates the decline in gait quality and physical function [24–26]. In addition, gait disturbance in the elderly may be an underlying cause of multilayered neurologic deficits or joint and skeletal system abnormalities [27,28]. Normal gait requires the dynamic integration of central and peripheral nervous systems acting on an intact musculoskeletal framework [29]. Poor gait is also one of the predictors of dementia [30], as cognitive impairment can disrupt gait control and result in gait disorder [31]. Therefore, identifying the most deteriorating gait parameters is vital for targeted and effective gait rehabilitation [32]. Additionally, osteoarthritis (OA) can lead to pain, stiffness, and gait dysfunction, but gait retraining could improve the symptoms [33]. To date, most research focused on gait problems of older adults with specific symptoms. There are fewer systematic reviews concerning the impact of Tai Chi intervention on people with gait impairment or how this condition influences their QoL. Therefore, this study systematically reviewed the relationship between Tai Chi exercise with postural stability and QoL in older adults with gait impairment by critically evaluating and summarizing the literature in Chinese and English language.

## 2 Methods

### 2.1 Eligibility criteria

The inclusion criteria for this review were developed according to the PICOS (Population, Intervention, Comparison, Outcome, and Study Design) principles (Table 1): 1) Research that utilizes a randomized controlled trial (RCT), 2) Tai Chi as an intervention method, including all types of Tai Chi. 3) Research subjects is ≥ 60 years old, and 4) Articles published in Chinese or English language. Meanwhile, studies that focused on participants < 60 years old were excluded.

### 2.2 Search strategy for the identification of studies

A total of six electronic databases were utilized for the literature search: PubMed, Scopus, Web of Science (WoS), CNKI, EBSCOhost (SPORT-Discus), and Google Scholar. In addition, the following keywords and Boolean operators were used in retrieving relevant articles: "Tai Ji" "Tai Chi" AND "Postural Balance*" "Postural Control*" AND "Gait disorders" "Locomotion Disorder*" "Ambulation Disorder*" "Gait Dysfunction*" "Unsteady Gait" AND "old people" OR "elderly" OR "senior*" OR "old adult*" OR "aged" OR "older people" OR "older adults" OR "geriatric". These keywords were used to search for other published systematic literature, while Mesh was utilized to complete the search in PubMed. Meanwhile, the search strategy used in the Scopus search bar was title/abstract/keywords and full-text or topical searches for other databases. Literature search dates are from inception to May 2023.

**Table 1. Inclusion criteria.**

| Items | Detail |
| --- | --- |
| Participants | Male or female, ≥60 years old, gait problems |
| Intervention | Tai Chi |
| Comparison | Control group, no exercise group or exercise group |
| Outcome | Balance, mobility, gait ability, strength, fall rate, quality of life (QoL) |
| Study designs | Randomized Controlled Trial |

## 2.3 Study selection

A total of 584 articles were found from six different databases, and 43 duplicates were later deleted using the Endnote (Clarivate, USA) literature management software (Fig 1). The remaining articles were excluded according to the title and keywords, and articles without keywords were excluded by reading the full text. Finally, relevant articles were reviewed and finalized to be included in the study.

## 2.4 Protocol and registration

The search strategy, data collection, and planned analysis protocol for this systematic review are registered with PROSPERO (CRD42022378411): https://www.crd.york.ac.uk/prospero/.

## 2.5 Quality assessment data extraction

The PEDro scale is a scale for evaluating the quality of clinical trials [34]. The scale consists of 11 items, ten of which focus on external validity, and was scored based on whether the criteria were met ("Yes" = 1 and "No" = 0), with a maximum possible score of 10 points. Generally, a PEDro score > 5 indicates a high-quality article, while a score less than 5 indicates poor quality (Maher et al., 2003). Two researchers completed each assessment independently in this study, and third-party arbitration was utilized in disagreements Table 2. Subsequently, two reviewers extracted the data independently according to the PICO principle. The final extracted data included (a) references, (b) participants, age, and gender, (c) intervention type, frequency, and duration, (d) sample features, and (e) primary outcomes.

# 3 Results

## 3.1 Study selection and characteristics

An initial search of 584 articles from electronic databases: PubMed (21), Scopus (18), Web of Science (45), EBSCOHOST (SPORT-Discus) (314), CNKI (174), Google Scholar (9), and Reference (3). After deleting the duplicates (43), 404 irrelevant articles were excluded according to the inclusion criteria. Another 108 were excluded based on the title and abstract: not related to postural balance and QoL (29), not related to gait (39), theoretical research (7), non-RCTs (10), and age ≤ 60 (7). Finally, 16 articles were included in the final analysis [15,35–49] (Fig 1).

## 3.2 Study quality assessment

The first item on the PEDro scale was excluded from the total score without affecting the internal validity of the article. The scores for all selected articles ranged between 4 and 8; two articles scored 4, and 14 articles scored ≥ 5. Therefore, the methodological quality of the selected articles is high, indicating the reliable outcomes of this review. In addition, the 16 articles met the eligibility criteria with between-group comparisons and random allocation, but only one study performed the blinding of participants, assessors, and therapists fit Table 2.

## 3.3 Characteristics of included studies

Based on the 16 included studies, the participants' characteristics were as follows: (1) Total sample size: 1408 subjects, (2) Gender: 13 studies included both genders [15,35,36,38–42,44–48], one article did not describe the participants' gender [43], two studies described women [37,49]. (3) Age: All studies included participants aged ≥ 60, where 11 focused on elderlies between 60–70 years old [15,35,36,39–41,43,45–47,49], and five included those > 70 [37,38,42,44,48], (4) Participants' health description: 10 articles studies the Parkinson's

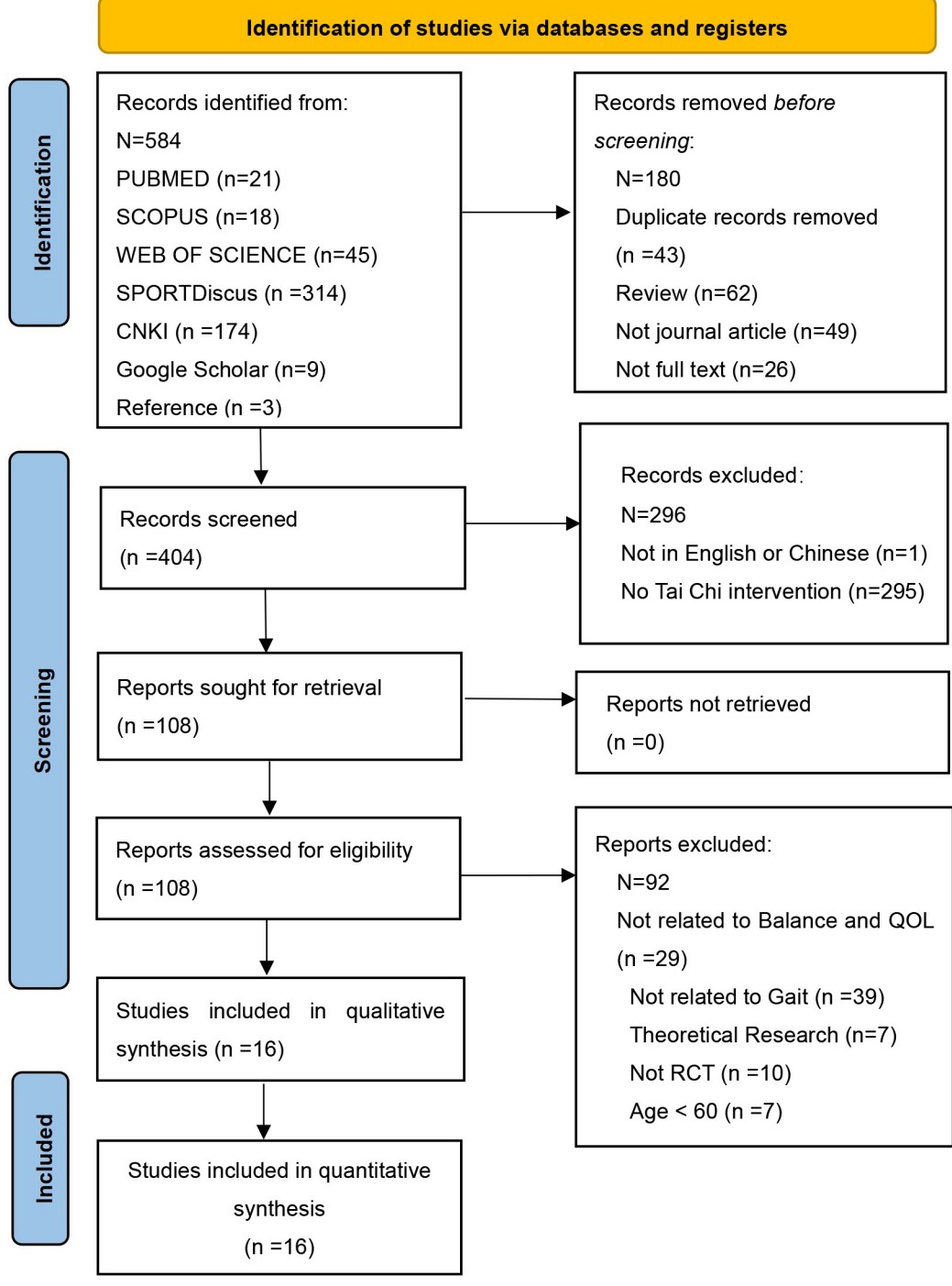

**Fig 1. PRISMA study selection process.**

population [15,35,39–41,43,45–48], two studies include elderlies that do not exercise [37,49], one study focused on the stroke patients [36], one study of elderlies with afraid or have a history of falls [42], one study included inactive participants [44], and one item studied those with mild cognitive impairment (MCI) [38] Table 3.

**Table 2. Summary of methodological quality assessment scores.**

| References | Eligibility criteria | Random allocation | Concealed allocation | Group similar at baseline | Blind subject | Blind therapist | Blind assessor | Follow-up | Intention-to-treat analysis | Between-group comparisons | Point measure and variability | PEDro score |
|---|---|---|---|---|---|---|---|---|---|---|---|---|
| Amano, Nocera [35] | 1 | 1 | 0 | 1 | 0 | 0 | 0 | 1 | 0 | 1 | 1 | 6 |
| Au-Yeung, Hui-Chan [36] | 1 | 1 | 0 | 1 | 0 | 0 | 0 | 0 | 1 | 1 | 1 | 6 |
| Li, Harmer [15] | 1 | 1 | 1 | 1 | 0 | 0 | 0 | 1 | 1 | 1 | 1 | 8 |
| Wei [49] | 1 | 1 | 1 | 1 | 0 | 0 | 0 | 0 | 1 | 1 | 1 | 7 |
| Kim, Je [37] | 1 | 1 | 1 | 0 | 1 | 1 | 1 | 0 | 1 | 1 | 1 | 9 |
| Fogarty, Murphy [38] | 1 | 1 | 0 | 0 | 0 | 0 | 0 | 0 | 1 | 1 | 1 | 5 |
| Hackney and Earhart [39] | 1 | 1 | 0 | 1 | 0 | 0 | 0 | 0 | 0 | 1 | 1 | 5 |
| Lin, Liling [40] | 1 | 1 | 0 | 0 | 0 | 0 | 0 | 1 | 1 | 1 | 1 | 6 |
| Li, Jianlan [41] | 1 | 1 | 0 | 1 | 0 | 0 | 0 | 0 | 1 | 1 | 1 | 6 |
| Chewning, Hallisy [42] | 1 | 1 | 0 | 1 | 0 | 0 | 0 | 1 | 1 | 1 | 1 | 7 |
| Choi, Garber [43] | 1 | 1 | 0 | 1 | 0 | 0 | 0 | 1 | 0 | 1 | 1 | 6 |
| Li, Harmer [44] | 1 | 1 | 0 | 1 | 0 | 0 | 0 | 0 | 1 | 1 | 1 | 6 |
| Yi, Jianxing [45] | 1 | 1 | 1 | 1 | 0 | 0 | 0 | 0 | 1 | 1 | 1 | 7 |
| Gao, Leung [46] | 1 | 1 | 1 | 1 | 0 | 0 | 0 | 1 | 1 | 1 | 1 | 8 |
| Zhang, Hu [47] | 1 | 1 | 1 | 1 | 0 | 0 | 0 | 1 | 1 | 1 | 1 | 8 |
| Khuzema, Brammatha [48] | 1 | 1 | 1 | 1 | 0 | 0 | 0 | 0 | 1 | 1 | 1 | 7 |
| **Total** | 16 | 16 | 7 | 13 | 1 | 1 | 1 | 7 | 13 | 16 | 16 | |

### 3.4 Intervention characteristics

Intervention characteristics included intervention type, duration, and frequency. All studies in this review utilized Tai Chi intervention, including two combined intervention studies [38,41]. Furthermore, seven items explicitly reported using the Yang style [35,39,40,42,44,46,47]: three studies chose the Yang style with 24-form movements [44,46,47], two studies used eight movements [35,40], one study used a 4-form movement [42], and one study did not specify the number of movements [39]. Meanwhile, two studies opted for Sun-style Tai Chi, particularly the 12 and 21 forms [36,37]. One study also utilized Taoist Tai Chi with 108 movements [38]. The remaining studies did not describe the Tai Chi routines, but a few of them stated the number of movements, including eight forms, 24 forms, six forms, and one form (cloud hands) [15,41,43,45,48,49].

Most studies were randomly divided into two groups for comparison (n = 14): experimental and control groups. Meanwhile, two studies randomly divided their subjects into two experimental groups and one control group. Only two studies used combined interventions [38,41]. Tai Chi intervention cycles were conducted between four to 24 weeks, two to five sessions weekly, with a duration of 30 to 90 minutes. The Tai Chi intervention time range in all articles varied from 4 to 24 weeks, but most articles used 12 weeks (n = 5) and 16 weeks (n = 3). Furthermore, most studies were conducted at 60 minutes per session (n = 12), followed by 90 minutes (n = 2) and 30–45 minutes (n = 2).

**Table 3. Sample features, main outcomes, and participants' characteristics.**

| References | Participants | Sample features | Intervention | Types of TC | Comparison | Outcomes |
|---|---|---|---|---|---|---|
| Amano, Nocera [35] | M/F N = 45 Age = 66–68 | PD | EG: Project 1: Tai Chi, Project 2: Tai Chi CG: Project 1: Qigong, Project 2: Non-contact control | Yang-style, first eight movements | EG: Project 1:16-Wk /2/ 60min CG: Project 1:16-Wk/2/ 60min EG: Project 2:16-Wk/3/ 60min CG: Project 2: No exercise | Gait (COP) ⟷, UPDRS III⟷ |
| Au-Yeung, Hui-Chan [36] | M/F N = 136 Age = 63.4 ± 10.7 | chronic stroke | EG: Tai Chi CG: breathing and stretching exercises | Sun style,12 forms | EG: 12-wk/ 1 hour of group practice was supplemented by 3 hours of self-practice. CG: similar to the Tai Chi group | Balance↑(COG), Mobility⟷ (TUG) Dynamic Standing Equilibrium↑ |
| Li, Harmer [15] | M/F N = 195 Age = 68± 9 | PD | EG: Tai Chi CG1: resistance training CG1: Stretching | eight-form | EG:24-wk/ 2/1 hour CG: like the Tai Chi group | Balance↑ (maximum excursion↑ directional control↑) Gait↑ (stride length↑ walking velocity↑), Mobility (TUG) Strength↑, UPDRS III↑, fall rate↓ |
| Wei [49] | F N = 48 Age = 60–70 | no exercise habit | EG: Tai Chi EG: Brisk Walking CG: no exercise intervention | 24-form | EG:16-wk/5/1 hour EG:16-wk/5/1 hour CG: no exercise intervention | balance ability↑ (SLO↑ SLC↑ SLS↑) |
| Kim, Je [37] | F N = 46 Age = 65–83 | no exercise habit | EG: Tai Chi CG: taekwondo | Sun-style, 21-form modified Sun-style | EG:12-wk/2/1 hour CG:12-wk/2/1 hour | balance↑ (FR↑, OLS↑) Gait ability↑ (gait velocity, step length, stride time, and cadence), Lower Strength↑ |
| Fogarty, Murphy [38] | M/F N = 40 Age = 71.55 (9.33) | MCI | EG: MIP Alone CG: MIP + TTC | Taoist Tai Chi, 108 movements | EG:10-wk/2/90 min CG:10-wk/2/90 min | Balance⟷ Gait ⟷ Cognition⟷, physical mobility⟷ |
| Hackney and Earhart [39] | M/F N = 33 Age = 64.9± 8.3, 62.6 ± 10.2 | PD | EG: Tai Chi CG: no exercise intervention | Yang Short Style of Cheng Man Ching | EG:13-wk/2/1 hour CG: no exercise intervention | balance↑ (BBS↑) gait↑ (tandem stance test↑ six-minute walk↑) functional mobility↑ (backward walking↑) UPDRSIII↑, satisfaction and well-being↑ |
| Lin, Liling [40] | M/F N = 80 Age = 62–83 | PD | EG: Tai Chi CG: not mentioned | Yang-style, eight movements | EG:16-wk/3/60min CG: not mentioned | Gait (gait velocity ↑, stride length↑, gait frequency⟷) UPDRS III↑ |
| Li, Jianlan [41] | M/F N = 80 Age = 64. 1 ± 8. 8,66. 9 ± 8. 5 | PD | EG: Medicine combined with tai chi cloud hands and traditional Chinese medicine fumigation CG: routine care | cloud hands, One movement | EG:8-wk/5/60min +Fumigate the affected limb, 15 min /1d CG: routine care | balance↑ (BBS) Gait (gait speed↑, stride length↑) |
| Chewning, Hallisy [42] | M/F N = 242 Age = 75.0 ±7.4,72.8 ±7.0 | Fear of falls/ history of falls | EG: Tai Chi CG: not mentioned | Yang-style, Fab Four tai chi movements | EG: EG:6-wk/2/90 min CG: no exercise intervention | balance↑ Gait↑(TUG), tandem or single leg balance) mobility ↑ strength↑ |
| Choi, Garber [43] | not mentioned N = 24 Age = 60.81 ±7.6,65.54±6.8 | PD | EG: Tai Chi CG: no exercise intervention | not mentioned | EG:12-wk/3/60 min CG: no exercise intervention | balance↑(one-leg standing test, tandem gait test), Gait↑(tandem gait test) UPDRS III↑ADL⟷ |
| Li, Harmer [44] | M/F N = 256 Age = 70–92 | Inactive | EG: Tai Chi CG: regular stretching | Yang-style, 24-form | EG:24-wk/3/60 min CG:24-wk/3/60 min | balance↑(BBS↑ Dynamic Gait Index↑ functional-reach↑ single-leg standing↑, fall rate↓ |

*(Continued)*

**Table 3.** (Continued)

| References | Participants | Sample features | Intervention | Types of TC | Comparison | Outcomes |
|---|---|---|---|---|---|---|
| Yi, Jianxing [45] | M/F N = 40 Age = 63.35 ±8.72,64.83±9.29 | PD | G1: Tai Chi G2: walking | Health Qigong 24 Forms Tai Chi | G1:4-week/5d/2/30-45min G2:4-week/5d/2/>40min | balance↑ (BBS↑), UPDRS III↑ |
| Gao, Leung [46] | M/F N = 76 Age = 69.54 ±7.32,68.28±8.53 | PD | EG: Tai Chi CG: no exercise intervention | Yang-style, 24-form | EG:12-wk/3/60 min CG: no exercise intervention | balance↑(BBS↑), UPDRS III⟷ |
| Zhang, Hu [47]) | M/F N = 40 Age = 66.00 ±11.8,64.35±10.53 | PD | G1: Tai Chi G2: Multimodal Exercise Training | Yang style, 24-posture short-form | EG:12-wk/2/60 min CG:12-wk/2/60 min | balance⟷ (BBS⟷,) Gait⟷ (stride length, gait velocity), UPDRS III |
| Khuzema, Brammatha [48] | M/F N = 27 Age = 60–85 | PD | EG: Tai Chi EG: Yoga CG: Conventional balance exercise | 6-form | EG:8-wk/5/30-40 min EG:8-wk/5/30-40 min CG:8-wk/5/40-45 min | balance↑(BBS↑) Functional mobility↑ (10-m walk↑ TUG↑) |

**Note:** sig↑; not sig↔; PD: Parkinson's disease; MCI: Mild cognitive impairment; SLS = single-leg stance; SLC = with eyes closed; SLO = with eyes open; OLS = one-leg standing; TUG = Timed Up and Go; FR = Functional Reach; ADL = activities of daily living; BBS = Berg Balance Scale; UPDRS III = the Unified Parkinson's Disease Rating Scale Motor Subscale 3; COG: Center of gravity; TC = Tai Chi; RCT = randomized controlled trial; WK = week; 6-MWD = 6-minute walking distance; TST = tandem stance test; MIP: Memory intervention program.

## 4 Outcomes

This section summarizes the findings from 16 selected articles that utilized Tai Chi as a training method and reported improvements in postural stability and QoL in older adults.

### 4.1 Effects of Tai Chi on postural balance

A total of 14 studies assessed the postural balance in older adults with different health conditions [15,36,37–39,41–49]:

- Eight studies evaluated the postural balance in the Parkinson's population.

- Two studies evaluated the postural balance ability of participants without exercise habits.

- One study evaluated the postural balance in patients with a fear of falls or a history of falls and chronic stroke.

The following measurements were used to assess postural balance: Seven studies used the Berg Balance Scale (BBS) scale, which is the most common scale for postural balance evaluation [39,41,44–48]. Most studies demonstrated improvements in the BBS score post-Tai Chi intervention, except one which reported no significant improvements [47]. Moreover, the single-leg stance test was reported in three studies on postural balance with significant improvements [37,43,44]. This study also found that the FR scale is sometimes used as an indicator of response postural balance ability [37,44], the final measurement has been significantly improved. while one article did not detail the test indicators used in their study [38]. Based on the findings of 14 studies, Tai Chi intervention between six to 24 weeks, two to five times weekly, with a duration of 30–90 minutes per session, significantly impacted postural balance among the elderly. In contrast, two studies reported no significant improvements in postural balance in their study population. Different styles of Tai Chi interventions may improve

postural balance ability. In this study, five items were Yang-style Tai Chi, and two items were Sun-style Tai Chi.

## 4.2 Gait analysis

Ten studies reported on gait performance [15,35,37–43,47]:

- seven studies assessed the gait of Parkinson's patients,

- one study evaluated the gait of a no-exercise population,

- one study observed the gait performance of the mild MCI population, and

- one study monitored the gait of participants with a fear of falls/history of falls.

There was more than one gait measurement method used in the selected studies, including walking velocity [15,37,40,41,47], stride length [15,40,47], step length [15,35,37,41,47], tandem stance test [39,43]. Parkinson's patient measurements include four using walking velocity [15,40,41,47], three using stride length [15,40,47], and four using step length [15,35,41,47]. Two Parkinson's population measurements use tandem stance tests [39,43]. The gait measurement of the no-exercise population is mainly to compare gait velocity, step length, stride time, and cadence before and after. The mild MCI population also focuses on gait indicators but does not specify the specific measurement indicators. The total intervention time was six to 24 weeks, two to five times weekly, and 30–90 minutes per session. Lin, Liling [40] and Zhang, Hu [47] exhibited that the gait of older adults improved significantly, but had no effect on stride frequency and length.

## 4.3 Mobility

Six studies reported mobility results Au-Yeung, Hui-Chan [36], Chewning, Hallisy [42] Li, Harmer [15], Fogarty, Murphy [38], Hackney and Earhart [39], Khuzema, Brammatha [48] that were measured using several methods, such as the TUG indicators (six studies) and other walking tests. Two functional mobility tests were included in the mobility tests in this study. The findings showed that two of the three items in the Parkinson's group were functional mobility [39,48], which measured backward walking, 10-m walking, and TUG. One in the chronic stroke group, one in the MCI group using TUG measurements, and one in fear of falls/history of falls group. The intervention time was six to 24 weeks, two to five times weekly, and 30–90 minutes per session. The mobility indicators illustrated that most Tai Chi treatment groups exhibited improvements in mobility scores, except for two studies that reported no significant enhancements [36,38].

## 4.4 Unified Parkinson's Disease Rating Scale (UPDRS) III

Eight studies utilized the UPDRS III indicator [15,35,39,40,43,45–47] to evaluate the impacts on elderly Parkinson's patients. Five studies assessed the reduction in motor ability and discovered improvements indicating that the motor capacity of the Parkinson's population has been improved. It is worth noting that four of the eight articles pointed out that Yang-style Tai Chi intervention was used, but the impact on the results was different. Two studies showed that Yang-style Tai Chi intervention can improve UPDRSⅢ [39,40]. In contrast, UPDRSⅢ was not significant after the other two Yang-style Tai Chi interventions, which may be related to the different intervention cycles, frequencies, and time per week [46,47]. Other interventions on UPDRS III did not specify which style of Tai Chi was used. The total intervention time is from 4 weeks to 24 weeks, 2 or 3 times a week, and the time is 30–60 minutes.

### 4.5 Strength

Two studies measure muscle strength in Fear of falls/history of falls and non-exercise populations [37,42]. Both studies focused on lower body muscle strength. These studies assessed lower body muscle strength using the 30-second chair stand [37,42] and found that lower limb strength was significantly enhanced after the Tai Chi intervention for six to 12 weeks, twice weekly, and 60–90 minutes per session. Two studies used different styles of tai chi interventions, one with Sun tai chi intervention and the other with Yang style tai chi but the final intervention effect of both types of tai chi was significant.

### 4.6 Fall rate

Two articles reported rates of falls were assessed in Parkinson's and Inactive populations, respectively [15,44]. One showed that the tai chi group had 67% fewer falls than the stretching group (incidence-rate ratio, 0.33; 95% CI, 0.16 to 0.71) [15]. Another study is not only about the rate of falls but also the risk of falls, the reduction of fear of falls has an improved effect [44]. One of the Tai Chi intervention styles is Yang style intervention and the other is not mentioned. The total intervention period was 24 weeks, two to three times weekly, and 60 minutes per session. The study findings revealed a significant reduction in fall incidence several weeks post-intervention.

### 4.7 Quality of Life (QoL)

Only two out of 16 studies reported how the Tai Chi intervention influenced the QoL of Parkinson's disease patients [39,43]. No mention was made of the quality of life of other groups of people. One study conducted a survey and revealed that the patients were relatively satisfied with their well-being(median = 2.0 (25%: 2.0, 75%: 3.0) [39]. Conversely, another study that assessed participants' QoL using the ADL scale reported that the Tai Chi intervention had no significant impact on their daily activities(7.91±1.81, 5.82±3.37, p = 0.119) [43]. The total intervention period was 12 to 13 weeks, twice to thrice weekly, and 60 minutes per session. In terms of intervention style, one of them used Yang style intervention and the final result was effective [39], and the other did not mention Tai Chi style [43].

## 5 Discussion

Gait and balance impairments are common forms of physical deterioration among older adults due to chronic inactivity [50,51]. Balance and impairments could reduce their independent mobility and increase the risk of falls and injuries [52]. A study showed that 32% of individuals over 75 experience gait and balance impairments [53]. Gait abnormalities are usually the result of illness or injury, but can also be caused by nerve damage, weak muscles, or joint problems [54]. Therefore, this review focused mainly on the impacts of Tai Chi intervention on postural control, gait ability, and QoL in older adults with gait problems and provides effective exercise programs for patients with different gait abnormalities and prevent gait deterioration.

The 16 selected articles utilized different types of Tai Chi as a study intervention, such as Yang-style, Sun-style, Taoist Tai Chi, and a few other unspecified forms. Seven studies in this review chose the Yang style, the most common form of Tai Chi intervention [55]. This Tai Chi form consists of simplified movements and postures, thus, an easy exercise for the elderly to learn and practice. Consequently, Tai Chi improved balance, gait, mobility, lower body strength, UPDRS III, and QoL among older adults. Despite that, several indicators were not significantly improved, two gait indicators are Yang-style Tai Chi intervention for 16 weeks, of

which 3 times a week for 1 hour can improve the gait of Parkinson's patients, while the other intervention is 2 times a week, every day Times 1 hour have no noticeable effect in people with Parkinson's disease. Therefore, this is attributed to differences in training frequency and duration [35,36,38,46,47].

Generally, the studies suggested that Tai Chi intervention improved postural stability (BBS, OLS, FR), gait ability (walking velocity, stride length, step length, tandem stance test), mobility (TUG, walking), motor function (UPDRS III), lower body strength (30-s chair stand), fall rate, and QoL (ADL) compared with no intervention or control group. Furthermore, Tai Chi benefited older adults with Parkinson's disease, stroke, MCI, fear or history of falls, physical inactivity, and no exercise habits to maintain their balance and gait stability. Moreover, gait and balance impairments were most common among Parkinson's disease patients (n = 10) in nine mixed-sex studies and one research with unspecified gender. Meanwhile, the subjects of the two non-exercise population studies were only women, and the remaining subjects were mixed genders. Therefore, no study focused solely on the male population.

Tai Chi can also improve balance among older adults, as demonstrated by the BBS, OLS, and FR indicators, the first two of which are ubiquitous balance outcome measurement tools [56]. Nevertheless, one study did not observe improvements in the participants' BBS index [47], while another reported no improvements without detailing the assessment tools. This is a combined intervention for people with mild cognitive impairment. The results of the study showed that Taoist Tai Chi combined with MIP intervention did not have a significant effect. The BBS index is widely used to evaluate the balance function, consisting of 14 items. The higher the final score, the better the balance ability [57]. Furthermore, 14 studies reported on balance, 12 of which demonstrated that the balance function of the Tai Chi intervention group was significantly higher than that of the control [15,36,39,41–44,46,48,49]. In contrast, two studies suggest that other interventions are necessary to improve balance functions among older adults [38,47]. Additionally, numerous studies suggested 12 weeks of training with a weekly program and 60–90 minutes per session is adequate for visible improvements. Finally, one study reported that a 10-week intervention had no significant effect on the balance among older adults [38].

Gait assessment is performed mainly on patients with neurological diseases. Furthermore, gait disturbances can be triggered by fatigue and proprioceptive deficits, and neuropathy limits proprioception and impede coordinated gait and constant speed maintenance [58]. In this review, the impacts of Tai Chi intervention on the stride index varied between the selected studies. For instance, the stride length improved significantly in three studies [15,37,40], but one study did not observe significant improvements in the participants' stride length [47], which may be due to multimodal exercise training in the control group the intervention effect is better than Tai Chi. Moreover, the studies utilized different walk tests to assess the participants' forms: five studies performed a step-length test [15,35,37,41,47], and all but one study showed significant improvements in step length. Another study assessed gait via walking velocity, the best indicator of gait function [59]. Meanwhile, one study assessed gait frequency, but did not report significant outcomes. The tandem stance test, consisting of two items, was also used as an effective gait test index and demonstrated significant and positive findings. Several articles also used TUG to measure gait and mobility in older adults [42]. In summary, Tai Chi intervention was carried out for 12–16 weeks, twice or thrice weekly, and 60 minutes per session could positively impact older adults, as supported by earlier [37,40,47,49].

Functional mobility is assessed using the TUG, a series of complex tasks with various elements that measure gait performance, balance, and mobility [60]. Twice-weekly training improved functional mobility in people with MCI, Parkinson's disease, chronic stroke, and fear of falls/history of falls populations, and the results were consistent across studies

[37,39,42,48,61]. Contrary to the > 12-week intervention, the 10-week intervention plan did not significantly affect mobility [38].

The UPDRS III is a popular test index to assess motor function in Tai Chi trials [62], which was used by eight studies to evaluate the performance among Parkinson's patients. Intervention duration for motor rating scales varied between studies, ranging from four to 24 weeks, with extended periods illustrating higher efficacy than shorter training. Notably, one trial showed significant improvement within four weeks of intervention [45], possibly due to the training frequency (five weekly sessions, twice a day). In a different study [46], the UPDRS III index decreased slightly after 12 weeks of intervention, but there was no significant difference.

Strength is one of the essential components of physical fitness, and it can be used as one of the indicators to measure health status [63]. Decreased postural stability can lead to lower extremity impairment, and there is a strong link between lower extremity function and balance [64]. Aging leads to muscle mass and strength loss, which could be prevented by building muscles to reduce bone density loss [65]. Only two out of the 16 selected studies evaluated leg strength among older adults. Overall, lower extremity strength improved significantly post-Tai Chi intervention, indicated by the 5 × Sit-Stand STS and 30-s STS tests [37], and 30-s chair stand [42]. The intervention period was between six to 12 weeks, twice weekly, and 60 or 90 minutes per session. Furthermore, no studies have reported upper extremity muscle strength in older adults.

Falls are a complex problem faced by older adults. Studies have found that Parkinson's disease patients are prone to potentially fatal falls [15,66]. Two studies recorded participants' fall rates over six months [15,44], and one study showed that the tai chi group had a 55% rate of falls after the intervention compared to 67% in the stretching group, so the tai chi group was more effective. Additionally, Tai Chi intervention may be superior to stretching exercises because the former emphasizes the transfer gravity center, thus, allowing the body to maintain stability and effectively enhance neuromuscular rehabilitation. In summary, 24 weeks of Tai Chi training, twice to thrice weekly and ≥ 60 minutes per session, could reduce falls in older adults.

The QoL improvement is a significant concept and the main target for research and practice in health and medicine [67]. Life satisfaction among older adults is linked to health and mortality as they age [68]. Stride length and pace are closely related to physical performance and QoL [69]. Keeping older adults healthy and in control of their daily activities may lead to a better QoL [70]. For instance, the well-being of Parkinson's disease patients was improved after the Tai Chi intervention [39]. Nonetheless, there were no significant improvements in the participants' daily activities in another study [43]. As only one indicator was assessed in previous studies, there is an urgent need to investigate other factors that could influence QoL among older adults. The results of this systematic review revealed the effects of different styles of Tai Chi intervention on improving postural balance, gait, mobility, UPDRS III, lower extremity muscle strength, and quality of life. Compared with the control group, most studies confirmed the positive effect of Tai Chi intervention on patients with gait disorders.

## 6 Study limitation

Overall, the efficacy and benefits of Tai Chi training for patients with abnormal gait were highlighted in this review. Nevertheless, several limitations have been identified as follows:

1. Most studies included participants of mixed genders. Only two focused solely on women, while none chose older men as their study subjects.

2. The intervention frequency and period were inconsistent. This study contains articles with low-frequency intervention but significant improvement and high-frequency intervention

without significant improvement. Therefore, it is difficult to determine which intervention frequency would be most impactful for the participants.

3. There are limited reports on QoL in the literature, so all reliability and convincing power could be low.

4. Two studies combined Tai Chi interventions with other treatments, thus, making it impossible to determine how much Tai Chi intervention influenced the outcome.

5. Studies on gait disorder people among Parkinson's disease patients are relatively common compared to other health conditions, such as chronic stroke, MCI, fear of falls/history of falls, inactivity, and no exercise habits.

6. Different types of gait impairments could increase the risk of falls. Numerous studies have found that training in balance, gait, and muscle strength is effective in reducing falls. Nevertheless, there are few analyzes of falls in people with gait disorders in this article, and only two mentions are related to falls.

## 7 Conclusion

Tai Chi is a promising intervention to improve postural balance, gait, mobility, and UPDRS III in people with Parkinson's disease, chronic stroke, fear or history of falls, no exercise habit, MCI, and inactive populations. However, few studies analyzed the effects of Tai Chi on strength, fall rates, and QoL in individuals with OA, thus, warranting further investigation. Additionally, most studies demonstrated that Yang-style Tai Chi could improve postural stability in older adults with different gait disorders. Various test indicators were utilized in assessing postural balance and gait variables, such as BBS, where high scores suggest significant improvements in participants' balance. Nonetheless, literature on the effects of Tai Chi on stride frequency and endurance remains scant. Meanwhile, UPDRS III scores indicated that short-cycle, high-frequency Tai Chi training (12 weeks, three times per week), and long-cycle low-frequency training (24 weeks, twice per week) could improve motor function in Parkinson's patients. Moreover, lower extremity weakness leads to impaired walking and postural instability, which increases the risk of falls and reduces the QoL. In addition, Tai Chi combined with specific intervention methods potentially enhances postural stability in the elderly; hence, worth exploring in future research. Finally, more randomized controlled trials should be conducted in older males to understand the difference in Tai Chi intervention among older males and females.

## Supporting information

**S1 Table. Detailed search strategy.**
(DOCX)

**S2 Table. PRISMA 2009 checklist.**
(DOCX)

## Acknowledgments

The author would like to thank Dr. Li Kaijie for his help in writing the structure of the article.

## Author Contributions

**Methodology:** Kim Geok Soh, Yoke Mun Chan, Xiao Rong Bai, Fengmeng Qi, Nuannuan Deng.

**Supervision:** Kim Geok Soh.

**Writing – original draft:** Fan Xu.

**Writing – review & editing:** Kim Geok Soh, Yoke Mun Chan.

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
