## [Decision Letter · Decision Letter 0]

9 May 2023

PONE-D-23-09135Effects of tai chi on postural balance and quality of life among the elderly with gait disorders: a systematic reviewPLOS ONE

Dear Dr. XU,

Thank you for submitting your manuscript to PLOS ONE. After careful consideration, we feel that it has merit but does not fully meet PLOS ONE’s publication criteria as it currently stands. Therefore, we invite you to submit a revised version of the manuscript that addresses the points raised during the review process.

ACADEMIC EDITOR:Dear Corresponding Author Reviewer 1 and Reviewer 2 evaluated you paper and it was appreciated, at the same time minor revisions are requested. Please, read carefully all the comments and prepare a proper answer for each one and submit again the paper. 

We look forward to receiving your revised manuscript.

Kind regards,

Luca Russo, Ph.D.

Academic Editor

PLOS ONE

Journal Requirements:

2. Thank you for submitting your work to PLOS ONE. PLOS' guidelines require systematic reviews to be comprehensive and therefore up-to-date (https://journals.plos.org/plosone/s/submission-guidelines#loc-systematic-reviews-and-meta-analyses). Specifically, we expect systematic review searches not to have been done earlier than one year before the time of submission. To this effect, please include in the Methods section the date of the most recent search of the literature review conducted for your study. Thank you for your attention to this request.

Reviewers' comments:

Reviewer's Responses to Questions

**Comments to the Author**

1. Is the manuscript technically sound, and do the data support the conclusions?

Reviewer #1: Yes

Reviewer #2: Yes

2. Has the statistical analysis been performed appropriately and rigorously? 

Reviewer #1: N/A

Reviewer #2: Yes

3. Have the authors made all data underlying the findings in their manuscript fully available?

Reviewer #1: Yes

Reviewer #2: Yes

4. Is the manuscript presented in an intelligible fashion and written in standard English?

Reviewer #1: Yes

Reviewer #2: Yes

5. Review Comments to the Author

Reviewer #1: For all authors,

thanks for sending the paper to this journal, to improve the quality of the paper I will be giving some suggestions:

The English writing of the article in general needs to be improved. for example some words like elderly and older adults have been quoted differently, I suggest you include just older adults and delete elderly. gait and balance problems can be gait and balance impairments. and line 50 and 60 and other words

line 21-24 Some confusion, difficult to understand, you need to add population, and then factors and measures

line 30: reduce falls . Did you have the numer of fallers and no fallers, so you cannot say reduce or prevent falls

you used measues to decrease the risk of falls

line 45-46: exercise is non drug ( no understand) and all patients (which patients)

Outcomes: Balance, mobility, gait ability, strength, fall rate, quality of life (QoL)

You need to explain better the outcomes, which measurements each one were used

as you mentioned before postural balance is different than balance

fall rate this is the number of falls or fallers and no fallers and you mentioned fear of falling that is different

The results: in relation to TUG you have inclued in all measurements

TUG is funcional mobility- general mobility

you included in postural balance, gait, mobility

dont think this is correct

line 219: Two studies evaluated strength in the elderly's fear of falls/history of falls and no exercise

I mentioned before fear isnt falls

what tipe of strength has been evaluated....

Discussion

236 needs the objectives before

249 explain WHY did not improve

251-243 defined measures, explain

as I mentioned before

264 Why no improvment, explain

278 explain

300 why strenght

309 e 310 is not clear about rate of falls, explain and add the numbers

314-321 end the paragraph say about the relation of the title of the paper

340 you did not analyze (is meta-analyses) and rate of falls

350 say numbers frequency, weeks is better to improve

Reviewer #2: Some minor revisions below:

- Line 13-31, abstract: Authors should report in full the content of the acronym when it first appears in the abstract.

- Lines 27-31, the conclusions of the abstract should be more consistent with the conclusions of the work and above all report the limitations observed in all the works including the comparability of the studies, including duration, frequency of intervention, and type of movements, style.

- Line 85, "Tai Chi as an intervention method" the authors should better clarify what types of interventions are considered with the term Tai Chi and what styles are included.

- Line 103, authors should consider to include a PRISMA Statement Checklist or similar as supplementary material

(http://www.prisma-statement.org/documents/PRISMA_2020_checklist.pdf).

6. PLOS authors have the option to publish the peer review history of their article (what does this mean?). If published, this will include your full peer review and any attached files.

Reviewer #1: **Yes: **Dr Larissa Donatoni da Silva

Reviewer #2: No

---

## [Author Response · Author response to Decision Letter 0]

19 May 2023

Many thanks to the reviewer for his suggestion on manuscript number PONE-D-23-09135. Based on these comments, we carefully revised the manuscript. We appreciate your thoughtful comments. Based on these comments and suggestions, we carefully revised the manuscript. You will find our point-by-point responses to reviewers' comments/questions in the 'Response to Reviewers' document.

---

## [Decision Letter · Decision Letter 1]

30 May 2023

Effects of tai chi on postural balance and quality of life among the elderly with gait disorders: a systematic review

PONE-D-23-09135R1

Dear Dr. XU,

We’re pleased to inform you that your manuscript has been judged scientifically suitable for publication and will be formally accepted for publication once it meets all outstanding technical requirements.

Kind regards,

Luca Russo, Ph.D.

Academic Editor

PLOS ONE

Additional Editor Comments (optional):

Congratulations for the paper it was really appreciated by the reviewers.

Reviewers' comments:

Reviewer's Responses to Questions

**Comments to the Author**

1. If the authors have adequately addressed your comments raised in a previous round of review and you feel that this manuscript is now acceptable for publication, you may indicate that here to bypass the “Comments to the Author” section, enter your conflict of interest statement in the “Confidential to Editor” section, and submit your "Accept" recommendation.

Reviewer #1: All comments have been addressed

Reviewer #2: All comments have been addressed

2. Is the manuscript technically sound, and do the data support the conclusions?

Reviewer #1: Yes

Reviewer #2: Yes

3. Has the statistical analysis been performed appropriately and rigorously? 

Reviewer #1: N/A

Reviewer #2: N/A

4. Have the authors made all data underlying the findings in their manuscript fully available?

Reviewer #1: Yes

Reviewer #2: Yes

5. Is the manuscript presented in an intelligible fashion and written in standard English?

Reviewer #1: Yes

Reviewer #2: Yes

6. Review Comments to the Author

Reviewer #1: I would like to Thanks all authors to Review and introduced the comments on the manuscript!!

It is a great manuscript!

Reviewer #2: Comments have been addressed and this manuscript appears improved and therefore suitable for acceptance.

7. PLOS authors have the option to publish the peer review history of their article (what does this mean?). If published, this will include your full peer review and any attached files.

Reviewer #1: **Yes: **Larissa Donatoni da Silva

Reviewer #2: No

---

## [Editor Report · Acceptance letter]

1 Jun 2023

PONE-D-23-09135R1 

Effects of tai chi on postural balance and quality of life among the elderly with gait disorders: a systematic review 

Dear Dr. XU:

I'm pleased to inform you that your manuscript has been deemed suitable for publication in PLOS ONE. Congratulations! Your manuscript is now with our production department. 

Kind regards, 

on behalf of

Dr. Luca Russo 

Academic Editor

PLOS ONE